# Detection of Defects in Reinforced Concrete Structures Using Ultrasonic Nondestructive Evaluation with Piezoceramic Transducers and the Time Reversal Method

**DOI:** 10.3390/s18124176

**Published:** 2018-11-28

**Authors:** Guoqi Zhao, Di Zhang, Lu Zhang, Ben Wang

**Affiliations:** Faculty of Civil Engineering and Mechanics, Jiangsu University, International Joint Research Center for Key Structural Health Management of High-End Equipment, Zhenjiang 212013, China; meihensyzd@163.com (D.Z.); zzzlu@outlook.com (L.Z.); runningcosnail@163.com (B.W.)

**Keywords:** piezoceramic transducer, ultrasonic wave, wave propagation, non-destructive evaluation, reinforced concrete, time-reversal-method, damage image

## Abstract

Reinforced concrete is of vital importance in many civil and industrial structural applications. The effective bonding between steel and concrete is the core guarantee of the safe operation of the structures. Corrosion or other interface debonding in steel-concrete is a typical failure mode during the long service period of the structures, which can severely reduce the load-bearing capacity. The Non-destructive Evaluation technique has been applied to civil engineering structures in recent years. This paper investigates the evaluation of reinforced concrete structures that have interface defects, including the cross-sectional loss and cracks, by using the piezoceramic induced ultrasonic wave and time reversal method. Ultrasonic wave is used as actuating wave to obtain the signals with defect information. Time reversal method is applied to localize and characterize defect along the interface of the steel-concrete and to image the defect through the cross-sectional scanning. Experiments were conducted to perform Nondestructive Evaluation by using six reinforced concrete components with different levels of defects. The invisible damages were made by the cutting part of the steel and embedding a table tennis ball inside concrete structures. The results show that the time reversed method can locate and evaluate the defects along the steel reinforced concrete, and the obtained defect images at the cross-section of the concrete structure are accurate.

## 1. Introduction

Concrete is the most widely used construction material in civil engineering due to its long service life and durability in harsh environments [1]. This material has an excellent mechanical and structural performance when working with steel reinforced concrete during its service life. However, it has important issues in terms of durability when the concrete structures are exposed to corrosive environments and harsh conditions. Corrosion-induced cracking in pre-stressed concrete elements has been one of the most dominant factors that cause the deterioration of concrete structures. Since most of the corrosion-induced cracks in their early age are invisible, the qualitative and quantitative estimation of the corrosion-induced damage is not applicable before surface cracks can be observed [2]. Corrosion is a major safety and economic concern for various industries [3,4]. As a result, the maintenance of existing structures is of equal significance as the construction of new ones. Some researchers analyzed the debonding defect between the Fiber Reinforced Concrete Plastic (FRP) and concrete with the embedded Lead Zireinforced Concreteonate Titanate (PZT) sensors which have a potential in real civil engineering applications, civil engineering [5,6,7], and mechanical engineering [8,9,10]. Piezoceramic transducers have been widely used to detect the defects in structures such as rock bolt monitoring [11] and gas pipelining [12,13,14,15], and to detect the interlayer slide and collapse [16].

The time reversal method (TRM) is applied to damage detection or source locations in different reinforced concrete fields. Prior time reversed method studies focus on filtering single events out of recordings of low signal to noise (S/N)-ratio [17] or on the spatial and temporal accuracy of single event localization [18]. The aim is to transform a specific method within exploration geophysics to non-destructive testing [19]. Time Reversal (TR) approaches are applied in the Medical and Earth Sciences [20], which has been applied in non-destructive evaluations to improve flaw detection [21,22]. The time-reversal imaging with the multiple signal classification method for the location of point targets developed within the framework of the Born approximation is generalized to incorporate multiple scattering between the targets [23]. Time reverse modelling of wave propagation in elastic media is applied to localize and characterize acoustic emission using a numerical concrete model. This method can also eliminate the dispersion effects of guided waves such as Lamb waves [13,24]. Time reversed method is used to quantitatively represent the grout presence in detecting the post-tensioning tendon [25,26]. An effective approach using the time reversal technique and the piezoceramic transducers transducer was described in this paper to monitor the loading status of pin-connected structures, which are widely used in the construction industry [27]. Time reversed method localization using this highly scattering material is feasible using the rotated staggered finite-difference method. The localization of acoustic emissions with a limited number of sensors and using effective elastic properties were studied [19,28].

On the other hand, researchers also apply the ultrasonic wave technique to detect defects in reinforced concrete. Ultrasonic wave velocity is applicable to concrete cast inside a form and is often used to detect flaws and evaluate the compressive strength of hardened concrete [29]. The damage detection procedure is based on the ultrasonic wave propagation technique [30]. When concentrating on the ultrasonic wave propagation in concrete, it is important to introduce the three-dimensional mesoscopic and microscopic structures because ultrasonic waves in solids are transmitted by mechanical interaction between the adjacent media. It was observed that these factors were in good agreement between the simulation and measurements [31]. The piezoelectric transducers are located on both ends of the specimen and the measurements are taken periodically during the incrementally increased loading [32]. Rectangular piezoelectric ceramic patches were attached at the exposed ends of the rebar to monitor the wave transmission along the rebar with and without simulated corrosion, which was introduced in the form of the partial removal of the material from the rebar [33]. Two types of piezoceramic transducers elements were used in the experiments. Piezoceramic transducers disks were attached on the outer surfaces of the concrete beams to observe wave propagation in the concrete before and after a four-point bending test, and rectangular piezoceramic transducers patches were attached at the exposed ends of the rebar to monitor wave transmission along the rebar with and without simulated corrosion in the form of partial material removal from the rebar [34]. Numerical and experimental results show that the undamaged layer thickness in a deteriorated concrete structure is measurable using pulse-echo measurements when the deterioration depth is larger than the wavelength [30]. piezoceramic transducers has been applied into practice as a new intelligent material for structural health monitoring [30]. Ultrasonic imaging method has also been used to identify the defect in concrete during freeze-thaw (F-T) cycles [35]. To give a conclusion of other related reinforced concrete, more work should be focused on the corrosion like defects which may lead to the interface defects along the interface of steel and concrete, as well as the location of the defects.

The purpose of this paper is to detect the interface defects of reinforced concrete structures and to obtain the cross-sectional defect image. By using piezoceramic transducers, the pre-set interface defects can be identified by using time reversed method damage index (*DI*) and the determination of the location of the defect can be executed by the wave velocity and the time-of-flight. Then the ultrasonic probes were located at the certain position of the concrete surface to image the defect. At the end of the paper, experiments of the time reversed method imaging of artificial defects inside the concrete structures were conducted and results demonstrated the accuracy of the proposed imaging method.

## 2. Detecting System

The incident waves and the time reversed method evaluation procedure should be carefully selected due to the different detecting structures and different defects situation. This section mainly discusses the wave selection in the media and the time reversed method procedures.

### 2.1. Wave Selection for Time Reversed Method

To investigate the capability of the piezoceramic transducers sensor and the ultrasonic probes for identifying defects along the interface of steel and concrete in reinforced concrete structures, the incident waves in UT are determined by the geometry and the material of the detecting samples. Table 1 shows the main material parameters and the geometry of the detecting reinforced concrete samples.

In Table 1, *d* represents the diameter of the steel, *a* represents the side length, *l* represents the length of the material.

(1) Stress wave equations in concrete structures

The defects are present inside of the concrete samples, and the ultrasonic probes are located at the outer surface of the experiment samples. P-waves and S-waves are the main efficient ultrasonic waves that propagate in this infinite solid. Based on the material parameters listed in Table 1, the theoretical velocities of P-wave *C_L_* and S-wave *C_T_* can be respectively calculated as,
(1)$$c_{L}^{2}=\frac{\lambda +2\mu }{\rho }=\frac{E\left(1-\mu \right)}{\rho \left(1+\mu \right)\left(1-2\mu \right)}$$
(2)$$c_{T}^{2}=\frac{\mu }{\rho }=\frac{E}{2\rho \left(1+\mu \right)}$$
where λ,μ represents the lame parameter of the material; ρ is the material diversity.

(2) Longitudinal mode wave selection in reinforced concrete structures

Guided waves propagate in cylindrical structures, such as pipes and cylindrical rods [36]. By generating a signal vertical to the radius, we can excite the longitudinal guided wave [37], whose displacement is mainly in the radius direction and is sensitive to the cross-sectional mass loss along the steel reinforced concrete and the cracks in the concrete. The longitudinal guided wave can be used to evaluate the damage along the interface between the steel reinforced concrete and the concrete.

According to the material parameters of the experiments in Table 1, the dispersion curve of the guided wave propagates in a steel rod embedded in concrete is shown in Figure 1 which was obtained in Disperse. The diameter of the steel rod is 18 mm and the concrete specimen is 40 mm. This paper adopts the L(0,1) mode which is only sensitive to radius changes and only to its displacement by the radius to detect the defects along the interface of the reinforced concrete structures.

Figure 1a was the group dispersion curve, in which the group velocity is related to the frequency and the initial velocity is less than 4500 m/s. The primary L(0,m) modes lead to the S-wave as the frequency increases. In this reinforced concrete, in order to eliminate the multi-modes and the dispersion in the propagation procedure, the centre frequency of the actuated signals should be identified from Figure 1a, and the cut-off frequency can be identified at around 50 kHz.

A 5-cycle burst signal was used as the excitation signal, as shown in Figure 2. This signal is described as,
(3)$$S(t)=A[H(t)-H(t-n/f_{c})]\times (1-\cos(2\pi f_{c}t/n))\sin2\pi f_{c}t$$
where *H*(t) is the Heaviside function, *A* is the amplitude, *f_c_* is the Central frequency and *n* is the number of the cycles of the signal.

Figure 2 shows the incident waves in time-domain and in frequency-domain in this reinforced concrete. The 5-cycle tune burst has the advantage of high energy intensity and narrow frequency band. Please note that the Central frequency is determined based on the material’s property of the specimen, as listed in Table 2, which also shows other properties of the incident wave. 

### 2.2. Steel Reinforced Concrete Corrosion Detecting System Based on Time Reversed Method

The time reversed method in the axial direction of the reinforced concrete specimen is divided into two parts. One is the forward step which is performed in the defected specimen, and the reversed step is simulated in the same geometry as the defected one but in the intact specimen. The principle of the time reversed method in a two-dimensional cylinder is illustrated in Figure 3, where the tone burst of 5-number cycles *S*(*t*) is applied to the transducer at A side (Step 1), activating a wave signal that is captured by the transducer at the B side as *R*(*t*) (Step 2); the captures signal is time-reversed in the time domain as *S*’(*t*) = *R*(−*t*)) (Step 3) and reapplied to B’ side which is an intact cylinder; then the wave signal at the A’ side is collected and reconstructed of the original signal *R*’(*t*).

### 2.3. Reinforced Concrete Detecting Method Based on Time Reversed Method

The wave propagation in the concrete is the body wave and the ultrasonic probe is used to actuate and receive signals on the surface of the concrete. In Figure 4, the 5-number cycles tone burst *S*(*t*) is applied to generate a signal at the concrete surface; then the signals are captured by transducers on the same side as the generated one and each side takes this step (Step 2); the obtained signals are reversed in the time domain as *S*’(*t*) = *R*(−*t*) (Step 3) and regenerated at the same side where it comes from; then the wave signals are collected again and reconstructed as the original signal *R*’(*t*).

When the wave propagates in concrete, the wave scatters when it encounters a defect and the scattered wave containing the damage information would be received by the transducers on the outside surface of the specimen.

### 2.4. Damage Index

The Damage Index (*DI*) can be defined using the characteristic of wave signals in the time domain. In the time domain, the changes between the input tone burst *S*(*t*) and the reconstructed signal after time reversal *R*’(*t*), and the correlation coefficient $\rho_{S,R}$ of the two signals can be defined in Equation (3) [38] and the damage index can be defined as,
(4)$$\rho_{S,R}=\frac{n\sum a_{i}b_{i}-\sum a_{i}\sum b_{i}}{\sqrt{\left[n\sum a_{i}^{2}-{\left(\sum a_{i}\right)}^{2}\right]}\sqrt{\left[n\sum b_{i}^{2}-{\left(\sum b_{i}\right)}^{2}\right]}}$$
(5)$$DI=1-\rho_{S,R}$$
where *a_i_* represents the actuated signal, *b_i_* represents the reconstructed directed signal, *n* is the wave element in the time-domain. In this way, the higher the *DI*, the greater the possibility of the existence of the damage in the wave path is evaluated.

### 2.5. Imaging Algorithm

When the *DI* value is beyond the threshold, the image based detection takes place. Since the location along the Z-direction of the reinforced concrete specimen can be calculated by the time-of-flight, the transmitters can be relocated at the four sides of the concrete cross-section. In order to image the defects inside the reinforced concrete specimen, the received signals can be reversed in the time-domain to regenerate in the intact samples in COMSOL. By observing the reversed wave propagation, the maximum of the displacement field can predict the defect location. In addition, one-side-time reversed method can identify the direction of the defect and time reversed method from four sides can relatively depict the boundary of the defect, as shown in Figure 4. In this paper, we image the defect inside the structure according to the following image condition [28],
(6)$$I_{TRM}=\underset{t\in \left[0,T\right]}{\max}\left\|u_{i}(x,y,t)\right\|$$
where *I_max_* represents the pixel in the detecting plane and *u_i_* represents the displacement field in the revered time domain. For every point in the detecting section, the displacement field in the time reversed method progress can locate the defect or the signal source during the time period.

### 2.6. The Overall Time Reversed Method Based Defect Detection Procedure

The severity and the location of the defect are unknown since the damage is inside the structure and the detection is divided into two steps, which are depicted in Figure 5. The first step is detecting the severity of the interfacial defects by calculating the *DI* value of each reinforced concrete specimen. If the *DI* value is larger than the threshold, then the reinforced concrete specimen is regarded as the defect sample.

## 3. Experimental Setup

### 3.1. Specimens and Transducers

Figure 6 shows the reinforced concrete specimens for the experiments. The material and the geometry are listed in Table 1. In Figure 6a, the six specimens are shown. In Figure 6b, the geometry and the cross-section of the defect are depicted. There are total six reinforced concrete specimens were cast to perform the UT based on time reversed method. One of the reinforced concrete specimens is intact and the other five specimens have defects. Each of the five specimens with a defect has a cut on the steel reinforced concrete, ranging from 1 mm to 5 mm with the increment of 1 mm and, at each defect location, there was a ping pong ball. The length of each defect is 40 mm, which is equal to the diameter of the ping pong ball.

Figure 7 shows the ultrasonic probe and the equipment used in the detecting system. Shown in Figure 7a are the ultrasonic probes that were placed on the outer surfaces of the reinforced concrete specimen to detect the inside defects. Figure 7b shows the piezoceramic transducers that were used in detecting the interface defects along the steel reinforced concrete and the concrete. The diameter of the round piezoceramic transducers disk transducer is 10 mm with a thickness of 1 mm, and the dimensions of the rectangular piezoceramic transducers are 20 mm × 2 mm × 1 mm. The central frequency ranges from 50 to 150 kHz.

### 3.2. Experimental Setup

In the first step of the detecting system, as shown in Figure 6, the piezoceramic transducers were attached to both end surfaces of the steel reinforced concrete. The piezoceramic transducers at Z = 0 mm, was used as the actuator and the one on the opposite side of the steel reinforced concrete at Z = 800 mm was used as the receiving sensor. The ultrasonic probe was coupled with grease to generate and received stress waves on the outer surfaces of the concrete specimen. The attachment of the piezoceramic transducers is shown in Figure 8. In Figure 8a, the piezoceramic transducers transmitters are glued at both sides of the steel to generate and receive the cylindrical signals that mainly propagate in the centre of the steel. Figure 8b shows the surface placement of the ultrasonic probes to evaluate the inside defect.

The overall experimental setup is shown in Figure 9a and the functional diagram is shown in Figure 9b. Ultrasonic waves were generated by the arbitrary waveform generator (33,220 A; 20 MHz) with a voltage amplifier (7602 M), and an oscilloscope (DS07054A) received the input signal and the output signal. After acquiring the signals, MATLAB was applied to reverse the signals in the time domain and regenerate into intact reinforced concrete component in COMSOL to reconstruct the signals. Then the *DI* value can be calculated by comparing the reconstructed directed wave and the input wave in the same period. Furthermore, the second detecting step was applied. The ultrasonic probe was placed on the outer surface of the concrete at Z = 250 mm.

Attention should be paid that to the fact that, in the experiments, the value of the amplifier should be the same each time to maintain the same experimental condition to eliminate errors in identifying the defect reflecting signal.

## 4. Results and Discussion

### 4.1. Results

We repeated the experiment more than three times to gain stable received signals. Figure 10 shows the received signals in the time-domain as one example of the UT along the z-axis of the reinforced concrete specimen. In Figure 10a, the received signal was reversed in the time domain as the regenerated signal, and in Figure 10b, the signal was received in the intact steel reinforced concrete to reconstruct the directed wave.

The reconstructed signals are calculated in Equations (4) and (5). The experimental results of the *DI* values are plotted in Figure 11, which shows that the *DI* value increases with the depth of the defect on the steel reinforced concrete. The horizontal axis represents the cross-section loss caused by defects and the vertical axis is the calculated *DI* value. The dots represent the *DI* value of each reinforced concrete specimen and the line represents the fitted result based on each dots. In this experiment, the defects were set along the interface of the steel-concrete and the defect depths were comparatively larger than the real structures. On the other hand, the *DI_o_* value is different for each structure since the proportion and the construction condition may differ for different structures.

The location of the defect can be identified by the time-of-flight (TOF) of the defect reflected signal. In this experiment, the defect in the six reinforced concrete samples was the same distance from one end surface of the steel rod so that the TOF of the defect was the same as 6.58×10^−5^ s The calculated location of the defect was 250.04 mm and the error was equal to zero.

Then the ultrasonic probe was placed on the outer surfaces of both the intact and defected specimens to compare the effects of the interface damage on the received signals. Figure 12 shows the received signals at the outer surface of the reinforced concrete specimen. In Figure 12a, the received signal is at the intact section of the reinforced concrete specimen and in Figure 12b, the received signal is at the damaged section of the reinforced concrete specimen. The second wave packet represents the reflection from the steel rod or the damage, which means the damage reflects more than the steel rod at both sides of the interface with the concrete. This feature can be applied in the time reversed method process to continue the imaging step.

Figure 13 shows the image based detecting results. Figure 13a plots the intact section of the concrete time reversed method image. Figure 13b depicts the hot pot at (50 mm, 50 mm), which reveals the damaged image of the reinforced concrete specimen. By comparing the two images, it is noted that where there is a hole, there is the hot spot. Additionally, the red area depicts the defect boundary.

### 4.2. Discussion of the Two Defects Inside the Concrete Section

In this section, two defects inside the concrete specimens without steel are discussed to confirm whether the time reversed method can image and locate the two defects using UT. In the simulation, we pre-set two signal sources reinforced concretes located at (55 mm, 50 mm) and (45 mm, 55 mm), respectively. There are seven sensors at each side of the concrete boundaries to receive signals and they are regenerated at each location to trace the signal sources reinforced concretes. The time reversed method process is the same as detecting concrete with one defect. The signals are reversed in the time domain to regenerate at each received place. In Figure 14, the time reversed method procedure was shown to identify the displacement in t ϵ [0, T] which the x-axis and y-axis represent the location in millimeter. In Figure 14a, the waves initially propagate at the 100 mm × 100 mm section at t = 1.681×10^−4^ s. In Figure 14b the wavefronts started to interfere, then a hot point appeared at (40 mm, 40 mm) in Figure 14c. In Figure 14d, there are two hot points located at around the pre-set signal source reinforced concrete situation.

From the above image, we can identify the displacement interference near the area of the defect location, which shows that the time reversed method can identify the two defects inside concrete structures. Furthermore, more work should be performed in experiments to verify the robustness of this method.

## 5. Conclusions

Monitoring reinforced concrete structures with ultrasonic waves based on time reversed method both from the steel ends and the concrete surface can evaluate the defect severity and can identify the location and the boundary of the inside defects. Due to the different boundary conditions of the detecting samples, the incident waves are carefully selected by the material parameters and the geometry of the samples. The proposed time reversed method detecting systems are adapted to perform the experiments. From the overall results and analyses, the following conclusions are made:The defect depths along the interface of steel and concrete can be evaluated by an L (0,1) ultrasonic guided wave actuated at one end of the steel based on time reversed method. The *DI* curve was linear to the cross-section loss along the interface of steel and concrete.After detecting the location of the defect, another time reversed method detecting system was performed at a certain “defected” boundary surface of the reinforced concrete samples to continue the time reversed method damage image. By comparing the intact cross-section and the defect cross-section, the time-reversal waves can locate the damage inside the reinforced concrete samples by focusing on the area of the damage.In the discussion part, the concrete without steel is simulated to apply the time reversed method to identify the two signal sources. The results showed that the two signal sources inside the concrete can also be localized using UT based on time reversed method. This work can help the engineer detect multiple damages in reinforced concrete structures if the damages are not very far from each other.

## Figures and Tables

**Figure 1 sensors-18-04176-f001:**
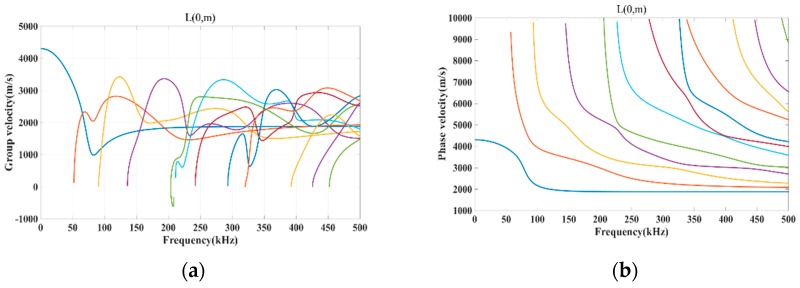
The dispersion curve of the concreteed concrete structure. (**a**) The group velocity dispersion curve (**b**) The phase velocity dispersion curve.

**Figure 2 sensors-18-04176-f002:**
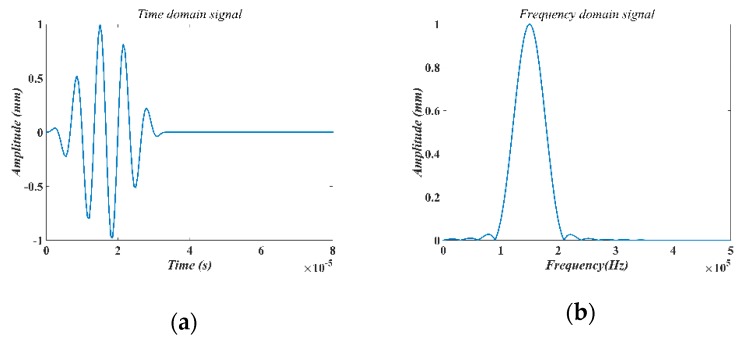
The incident wave. (**a**) Time-domain signal. (**b**) Frequency-domain signal.

**Figure 3 sensors-18-04176-f003:**
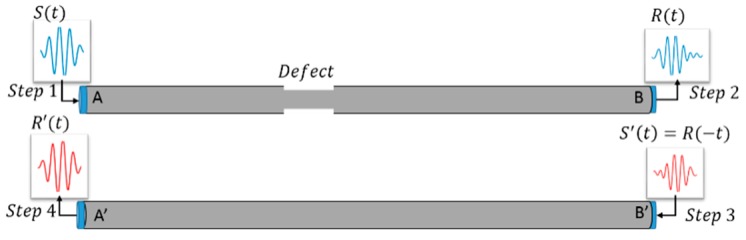
Steel time reversal detecting system.

**Figure 4 sensors-18-04176-f004:**
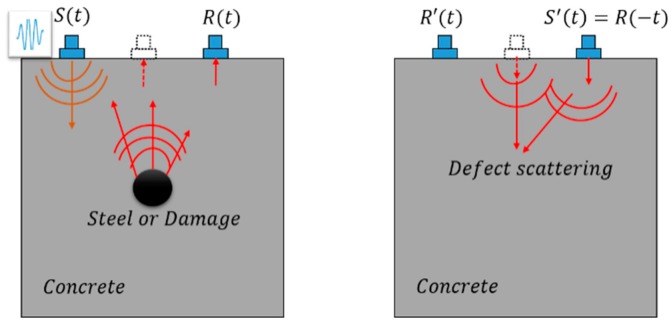
The concrete time reversed method detecting system.

**Figure 5 sensors-18-04176-f005:**
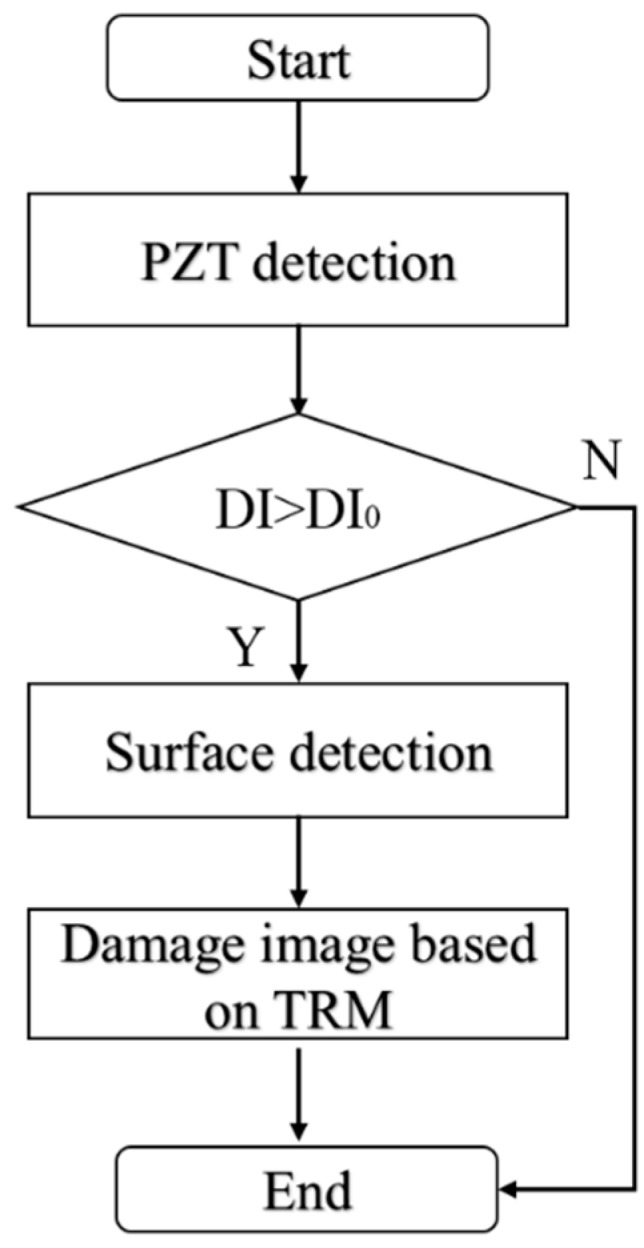
The time reversed method detecting procedure of reinforced concrete component.

**Figure 6 sensors-18-04176-f006:**
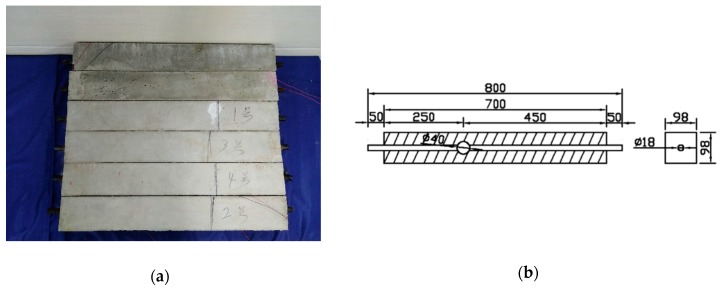
The detecting concrete sample. (**a**) The physical feature of reinforced concrete specimens. (**b**) 3D transparent feature of the sample (mm).

**Figure 7 sensors-18-04176-f007:**
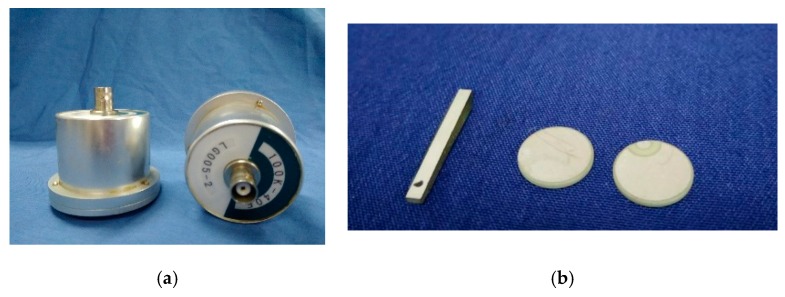
The transducers used in the detecting system. (**a**) Ultrasonic probe. (**b**) piezoceramic transducers.

**Figure 8 sensors-18-04176-f008:**
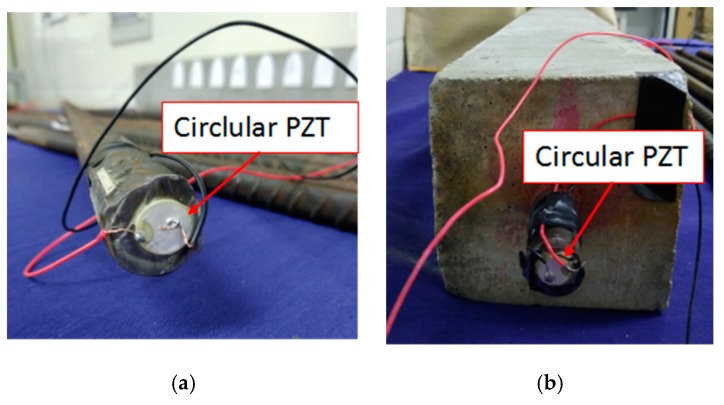
Piezoceramic transducers on the steel reinforced concrete. (**a**) Feature of the piezoceramic transducers attachment. (**b**) Placement of the ultrasonic probe.

**Figure 9 sensors-18-04176-f009:**
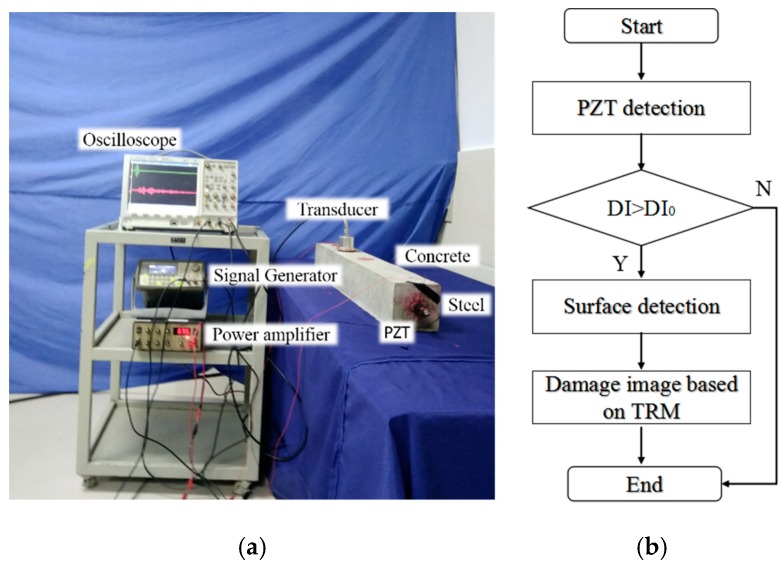
The concrete detecting system. (**a**) Experimental setup. (**b**) Functional diagram.

**Figure 10 sensors-18-04176-f010:**
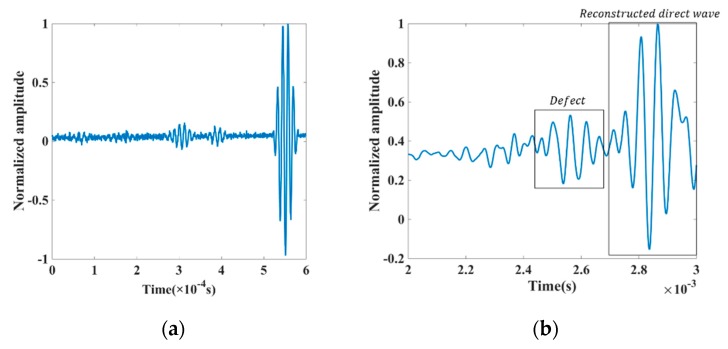
The signal reserved in the time domain in piezoceramic transducers. (**a**) The time-reversal signals (**b**) Results of the reversal progress.

**Figure 11 sensors-18-04176-f011:**
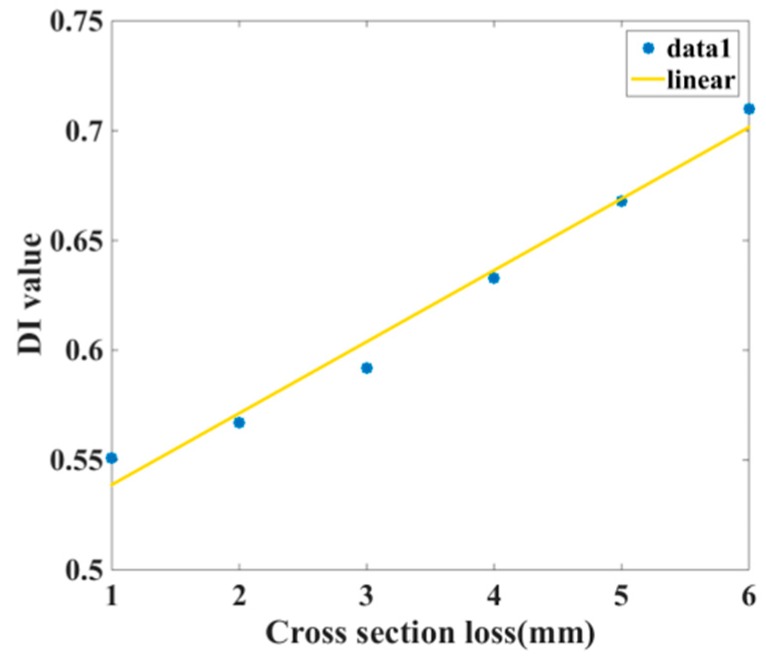
The *DI* evaluation of the reinforced concrete structures.

**Figure 12 sensors-18-04176-f012:**
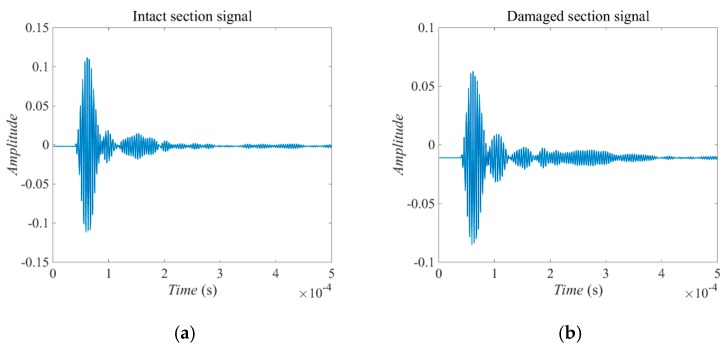
The received signals by piezoceramic transducers. (**a**) Signal at the intact section. (**b**) Signal at the damaged section.

**Figure 13 sensors-18-04176-f013:**
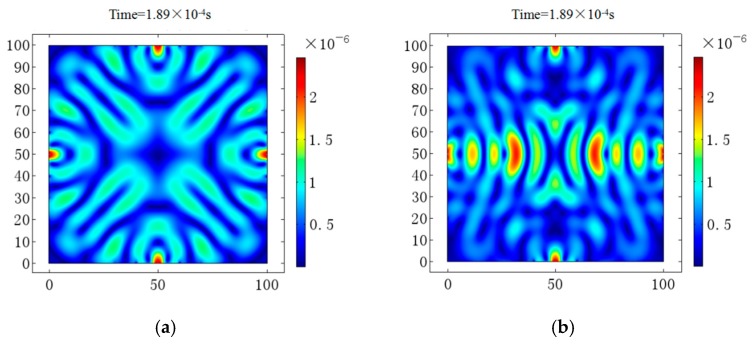
Imaging results. (**a**) Intact section image. (**b**) Damaged section image.

**Figure 14 sensors-18-04176-f014:**
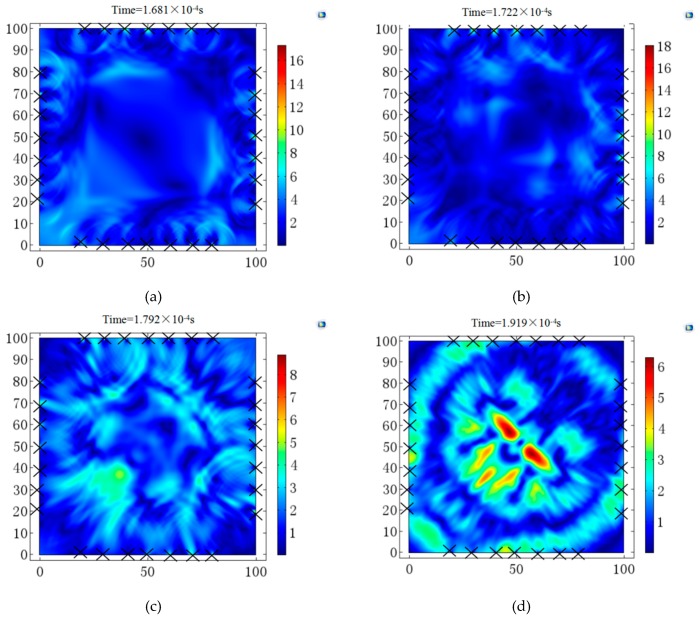
Time reversed method process of two defects at the cross section of concrete. (**a**) Image at T=1.681×10^−4^ s. (**b**) Image at T=1.722×10^−4^ s. (**c**) Image at T=1.792×10^−4^ s. (**d**) Image at T=1.919×10^−4^ s.

**Table 1 sensors-18-04176-t001:** The main parameters and geometry of the detecting samples.

Material	Steel	Concrete
Young modulus	200 GPa	25 GPa
Density	7850 kg/m^3^	2300 kg/m^3^
Poisson ratio	0.33	0.3
Geometry	*d =* 18 mm, *l =* 800 mm	*a =* 98 mm, *l =* 700 mm

**Table 2 sensors-18-04176-t002:** The parameters of the incident waves.

Material	Steel	Reinforced Concrete	Concrete
Central frequency	75 kHz	35 kHz	100 kHz
Amplitude	100	100	100
Number of cycles	5	5	5
Sampling frequencies	10 MHz	10 MHz	10 MHz

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
