# Peer review of "Detection of Defects in Reinforced Concrete Structures Using Ultrasonic Nondestructive Evaluation with Piezoceramic Transducers and the Time Reversal Method"

_sensors, 2018, doi:10.3390/s18124176_

Reviewer 1 Report

The overall message of the paper is well conveyed across the several sections of the paper. More than one section requires editing to fix grammar or spelling mistakes. These are my recommendations:

1.  I’d recommend avoiding all acronyms in the abstract

2.  At line 20, piezoceramically is not a word, piezoelectrically should be used instead

3.  Spell out FRP at line 43

4.  Spell out FIT at line 73

5.  Completely rewrite the paragraph from line 149 through 155, it’s hard to follow

6.  Line 230, severity is improperly used

7.  Minor language fixes are required in the conclusion paragraph, overall the message can be understood and well summarizes the content of the paper

Author Response

Dear reviewer,

Thank you very much to take your time to check this paper, and we have discussed about your recommendations.
Here are the answers of your recommendations:

1. We have avoid all the acronyms in the abstract;
2. We change the piezoceramically to piezoelectrically in line 20;
3. FRP is spelled out as Fiber Reinforced Plastic;
4. This sentence with “FIT” is deleted;
5. We rewrote line 149 to line 155;
6. We changed the severity to defect depth;
7. We checked the conclusion and changed some grammar errors.

Thank you very much for your kind recommendation.

Best regards!
Di Zhang

Reviewer 2 Report

General comments The manuscript deals with detection of internal defects in reinforce concrete structure using ultrasonic testing with PZTs and supporting by numerical modeling. The manuscript is prepared in a clear and concise manner, with a good literature review on all aspects of the manuscript, and with a clearly formulated goal of the study. Then the authors describe the measurement techniques used in their study and damage evaluation techniques and algorithms. Next. The authors describe specimens and experimental setup. Finally, the authors presented the results of damage detection and imaging following the proposed method. Although the paper is interesting, several corrections need to be introduced before publication. Detailed comments 1. To emphasize the novelty of a contribution I suggest to extend the literature review by discussing similar studies before formulation the goal of the presented study. 2. Some grammar formulations need to be corrected, e.g. “diameter of the steel” – steel has no diameter, but the reinforcing rod has. Moreover, numerous grammar errors are present. Careful review by the authors before sending the revised version is necessary. 3. The symbols of type of “L(0,m)” are not described enough, please extend the description. 4. The applied generator and amplifier should be also described in text with their most important parameters regarding the experiment. 5. Please describe data and fitting curve properly on Figure 11.

Author Response

Dear reviewer,

Thank you very much to take your time to check this paper, and we have discussed about your recommendations.
Here are the answers of your recommendations:

1. We have extended the literature review just before we introduced what we did in our paper;
2. We have checked the paper and changed some grammar errors;
3. We gave more explanation as we first introduced L(0,m) in this paper;
4. The sensors’ parameters were added in the text;
5. We have added some explanation of Figure 11. 

Thank you very much for your kind recommendation.

Best regards!
Di Zhang

Reviewer 3 Report

This experimental work is interesting in the terms of the application of ultrasonic non-destructive testing in construction engineering. The most interesting part of the work is the cross-sectional images of the reinforced concrete structures. However, there are lots of presentation flaws in the manuscript. Also, the uniqueness of the work, as is presented now, is not evident.

The English quality is below any threshold almost in every paragraph. It is very difficult to follow the story and understand what authors do want to tell to the reader.

The introduction is too general. The problems raised are obvious, known and being researched for many decades. Authors put their efforts and made an extended review of previous work, however at the end of the introduction they must provide a clear answer to the question, what are the unique and still unsolved scientific problems they are addressing in the present work. Also, I would avoid stressing the importance of using PZT transducers since other types of ultrasonic transducers may also be perfectly suited for the given application.

The heading of the 2-nd section in which authors present their methods has to be named adequately. "Detecting system" is not adequate and misleading.

Authors have to explain better how they obtain the dispersion functions presented in the Fig.1, since they do not provide any description of the relevant model nor the references.

Fig. 2 also needs a better explanation - is it a model of a signal or measured? And how it is related to the parameters in Table 2, where frequencies of 75, 35 and 100 kHz are listed? The center frequency of the signal in Fig. 2 is approximately 150 kHz.

When explaining the imaging part (from line 167) authors skip the adequate explanation of the image geometry. For example, they do not explain the direction of the Z axis. The poor explanation of the geometry leads to more unclear statements, such as at line 216.

Description of experimental setup has to be complemented with the functional diagram of the measurement system, possibly replacing the heavy packing of the manuscript with the photographs.

The heading of the Fig. 13 is not adequate.

There are many inadequate, underexplained and unclear terms used in the manuscript, such as in the lines 60-62; 68; 70-71; 73-78; 116-118; 125; 140 - 142; 149 - 155; 164; 177; 216; 217-219. Note, there are many other places in the manuscript with unclear sentences. Some of them possibly result from poor English, and some are due to the lack of supporting explanations.

Author Response

Dear reviewer,

Thank you very much to take your time to check this paper, and we have discussed about your recommendations. We have checked the paper and improve the language and revised some grammar errors.
Here are the answers of your recommendations:

1. We have changed the “detecting system” to “detecting method”;
2. We gave the parameters of the dispersion curve and the curve was obtained in Disperse;
3. We have given more description of Figure 2, this is used both in model and in experiments; Table 2 is what we did with different structures, and the Figure 2 is just an example of the central frequency 150kHz, to different central frequency, the time-domain and the frequency-domain can be different;
4. Actually, we have detect the Z axis in the first detecting step, and the second part only focus on the location at a certain cross section;
5. The experiment equipment function and the measurement system are both given in the paper;
6. We have rewritten the heading of Figure 13 Imaging results;
7. Other unclear terms are given more explanations in the text.

Thank you very much for your kind recommendation.
Best regards!

Di Zhang
